# Detectionof Ocean Internal Waves Based on Modified Deep Convolutional Generative Adversarial Network and WaveNet in Moderate Resolution Imaging Spectroradiometer Images

Zhongyi Jiang [1], Xing Gao [1], Lin Shi [1], Ning Li [1] and Ling Zou [1,2,*]

1   School of Computer and Artificial Intelligence, Changzhou University, Changzhou 213164, China; jiangzhongyi2008@hotmail.com (Z.J.); gaoxingtt@outlook.com (X.G.); slcczu@cczu.edu.cn (L.S.); lncczu@cczu.edu.cn (N.L.)
2   School of Microelectronics and Control Engineering, Changzhou University, Changzhou 213164, China
*   Correspondence: zouling@cczu.edu.cn

**Abstract:** The generation and propagation of internal waves in the ocean are a common phenomenon that plays a pivotal role in the transport of mass, momentum, and energy, as well as in global climate change. Internal waves serve as a critical component of oceanic processes, contributing to the redistribution of heat and nutrients in the ocean, which, in turn, has implications for global climate regulation. However, the automatic identification of internal waves in oceanic regions from remote sensing images has presented a significant challenge. In this research paper, we address this challenge by designing a data augmentation approach grounded in a modified deep convolutional generative adversarial network (DCGAN) to enrich MODIS remote sensing image data for the automated detection of internal waves in the ocean. Utilizing t-distributed stochastic neighbor embedding (t-SNE) technology, we demonstrate that the feature distribution of the images produced by the modified DCGAN closely resembles that of the original images. By using t-SNE dimensionality reduction technology to map high-dimensional remote sensing data into a two-dimensional space, we can better understand, visualize, and analyze the quality of data generated by the modified DCGAN. The images generated by the modified DCGAN not only expand the dataset's size but also exhibit diverse characteristics, enhancing the model's generalization performance. Furthermore, we have developed a deep neural network named "WaveNet," which incorporates a channel-wise attention mechanism to effectively handle complex remote sensing images, resulting in high classification accuracy and robustness. It is important to note that this study has limitations, such as the reliance on specific remote sensing data sources and the need for further validation across various oceanic regions. These limitations are essential to consider in the broader context of oceanic research and remote sensing applications. We initially pre-train WaveNet using the EuroSAT remote sensing dataset and subsequently employ it to identify internal waves in MODIS remote sensing images. Experiments show the highest average recognition accuracy achieved is an impressive 98.625%. When compared to traditional data augmentation training sets, utilizing the training set generated by the modified DCGAN leads to a 5.437% enhancement in WaveNet's recognition rate.

**Keywords:** MODIS; internal waves; classification; DCGAN; transfer learning; deep neural network; attention

## 1. Introduction

### 1.1. Internal Waves

Internal waves are a phenomenon arising from variations in temperature and salinity within the ocean, typically occurring in regions with density stratification [1]. These waves can exhibit significant amplitudes, exceeding 100 m, and travel distances spanning tens to hundreds of kilometers [2], rendering their detection challenging. Consequently, internal waves have emerged as a prominent focus of research within the field of oceanography.

The generation and propagation of internal waves are pervasive phenomena in the ocean, playing a crucial role in the transport of mass, momentum, and energy within oceanic systems. Breaking oceanic internal waves induces turbulent mixing, which in turn facilitates the vertical transport of water, heat, and other crucial climatic tracers within the ocean. This process holds significant importance as it actively influences the circulation patterns and the distribution of heat and carbon in the climate system [3], contributing to global climate change [4]. As a result, they can exert substantial influence on the safety and efficiency of marine engineering, oceanic communications, and oil exploration [5], and contribute to broader environmental factors, including their role in global climate change. Therefore, the study of internal waves in the ocean, particularly through the automatic recognition of these waves in remote sensing imagery, holds immense academic and practical significance.

With the continuous advancement of remote sensing technology, the exploration of internal waves in the ocean has shifted away from traditional field observations. Instead, researchers have embraced the use of remote sensing imagery, presenting a novel approach to this study. Among the various remote sensing instruments, the moderate resolution imaging spectroradiometer (MODIS) stands out as one of the most vital and distinctive tools currently available. It is integrated into platforms like Terra and Aqua and represents a state-of-the-art "all-in-one" optical remote sensing device in today's world.

MODIS boasts an impressive array of data with 36 bands, offering varying spatial resolutions of 250 m, 500 m, and 1000 m. Its scanning capacity spans an impressive 2330 km [6]. On average, local data can be acquired on a daily basis, and these data are readily accessible, making them the premier data source for global monitoring purposes. Nonetheless, the data provided by MODIS images are not only extensive in quantity but also substantial in size. Given that internal waves within the ocean occupy only a small fraction of the image area, the initial challenge lies in identifying and isolating images containing internal waves from the vast MODIS dataset before embarking on the process of detection and characterization.

### 1.2. Challenges and Research Objectives

To understand the characteristics of internal waves in MODIS satellite imagery, it is important to consider their distinctive optical signatures. When examining internal waves in the ocean through MODIS satellite imagery, they are frequently observed to be more prevalent in the vicinity of optical glare areas. This phenomenon can be attributed to the fact that signals received in these regions primarily result from sunlight reflecting off the sea surface. The presence of internal waves induces alterations in the sea surface roughness, leading to variations in the gradient of the sea surface at various scales. These variations, in turn, modulate the intensity of reflected sunlight received by remote sensors. Consequently, internal waves manifest as stripes with varying degrees of brightness in optical images. In visible satellite images, the divergent regions within the glare area typically appear as dark stripes, while the convergent regions appear as bright stripes. Outside the glare area, the divergent areas are characterized by bright stripes, while the convergent areas exhibit dark stripes. For a visual representation of how internal waves are depicted in MODIS imagery, please consult Figure 1.

The classification of remote sensing images depicting oceanic internal waves presents two significant challenges. Firstly, owing to the high resolution and abundant spatial and semantic information within remote sensing images, traditional machine learning methods like SVM, KNN, and decision trees struggle to effectively capture the intricate features inherent to these images. Secondly, the task of cropping and annotating remote sensing images is exceptionally labor-intensive, rendering it challenging for researchers to amass a substantial quantity of labeled image datasets for remote sensing data.

This study aims to address these challenges by developing an innovative approach to automatically detect and classify internal waves in MODIS satellite imagery, leveraging advanced deep learning techniques and data augmentation methods. By doing so, we aim

to contribute to the field of oceanography and remote sensing by providing a more efficient and accurate means of studying and monitoring internal waves in the ocean, ultimately enhancing our understanding of their role in oceanic and global climate processes.

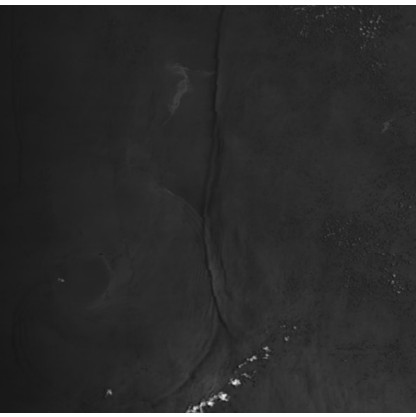

**Figure 1.** Oceanic internal waves in MODIS.

### 1.3. Contributions

The primary objective of this paper is to introduce an automated deep learning approach for the identification of internal waves within MODIS images.

1. We devised a modified DCGAN specifically tailored for data augmentation of MODIS remote sensing images.

2. We developed WaveNet, which incorporates a channel-wise attention mechanism, with the purpose of identifying internal waves.

3. We established a transfer learning methodology for the pre-training of WaveNet.

### 1.4. Paper Structure

The structure of this paper is organized as follows: Section 2 presents an overview of related work. The proposed approach is detailed in Section 3. Section 4 outlines the experimental methodology and reports the obtained results. Finally, in Section 5, conclusions and discuss possible avenues for future research are presented.

## 2. Related Work

Previous classification models, such as neural networks and support vector machines (SVMs) [7], typically featured a network structure with either one hidden layer node or none at all, earning them the designation of "shallow" classification models. However, these shallow classification models are categorized under shallow learning and possess limited capability to extract deeper features from constrained datasets, thereby constraining their overall model generalization ability [8].

In contrast, deep learning, as a novel machine learning paradigm, aspires to emulate the analytical learning capabilities of the human brain. By leveraging substantial volumes of training data and employing deep models with multiple hidden layers, deep learning can uncover more valuable features, consequently enhancing classification accuracy. Unlike shallow learning, deep learning architectures are characterized by their depth, typically consisting of more than three layers of hidden nodes. This depth enables them to explore deeper and more abstract features, thereby acquiring more precise feature information and ultimately affording superior generalization capabilities. In recent years, deep learning has achieved remarkable success in various image classification applications, prompting an increasing number of researchers to apply it to the domain of remote sensing image processing.

Deep learning has been integrated into the classification of hyperspectral data to harness the wealth of spectral information within hyperspectral images [1]. Notably, Haut et al. [9] introduced an innovative classification model that leverages both spectral and

spatial information present in hyperspectral data. This approach effectively mitigates the issue of rapid overfitting and accuracy degradation typically encountered when using convolutional neural networks (CNNs) with limited training data.

Bao et al. [10] employed the faster R-CNN framework, incorporating convolutional neural network features, to detect oceanic internal waves. Their efforts resulted in an impressive recognition rate of 94.78%. On a related note, Yu et al. [11] harnessed the lightweight convolutional neural network MobileNetv2 to extract deep and abstract image features. By combining feature fusion with bilinear pooling, they achieved higher accuracy in the realm of remote sensing image classification while utilizing fewer parameters and computational resources, surpassing other state-of-the-art methods.

In scenarios involving small-sample datasets, Li et al. [12] introduced a novel fault-tolerant deep learning approach known as RSSC-ETDL for remote sensing image scene classification. This method effectively mitigates the adverse effects stemming from inaccurately labeled datasets.

In recent years, researchers have also proposed many excellent models. In 2022, Zheng et al. [13] proposed a stripe segmentation algorithm based on SegNet for synthetic aperture radar (SAR) images. This method effectively identifies the presence of oceanic internal waves in SAR images and accurately locates both light and dark stripes associated with these waves. Also using SAR images, Tao et al. [14] construct a comprehensive dataset of 390 Sentinel-1 synthetic aperture radar (SAR) images, spanning multiple oceanic regions. These images are used to develop a machine learning model achieving high precision and recall when applied to detect internal waves (IW) across different scales and propagation directions in SAR imagery. Also in that year, Serebryany et al. [15] conducted an analysis using a collection of optical multispectral satellite images, including Sentinel-2 and Landsat-8 data, in conjunction with sea-truth data to identify internal wave features within the Black Sea.

Deep learning models typically necessitate multiple iterations of data analysis and processing, often involving substantial amounts of data [16]. Although the above work also has high performance in identifying ocean internal waves, it either requires a large number of high-quality datasets as data support, or the network model has room for improvement. However, the challenges associated with image cropping, annotation, and the acquisition of rare remote sensing images can present substantial obstacles for researchers when striving to compile extensive remote sensing image datasets during the data collection phase. Therefore, there is a special need in the field for a method that can greatly increase the data volume of remote sensing datasets, thereby effectively reducing the cost of annotation, and at the same time have a very high recognition rate of ocean internal waves.

This paper conducts classification on full-space images acquired through the Moderate Resolution Imaging Spectroradiometer (MODIS), encompassing four distinct categories: ocean scenes, clouds, terrestrial landscapes, and ocean waves. To address the challenge of limited dataset availability, the author employs an enhanced deep convolutional generative adversarial network (DCGAN) to substantially augment the data within each category sample. Additionally, a novel residual network is designed, taking into consideration the channel information of deep features, termed "WaveNet," to enable automated detection of internal ocean waves in MODIS images through an end-to-end approach.

## 3. Methods and Data

This section will introduce in detail the collection of remote sensing data, the construction of datasets, the modified DCGAN model structure and the WaveNet model structure. The entire workflow diagram is shown in Figure 2.

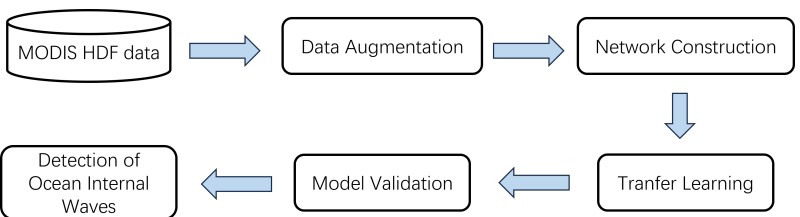

**Figure 2.** Flow chart of internal waves detection framework.

*3.1. Data Augmentation*

3.1.1. MODIS Images

First, obtain the HDF format data of MOD02QKM from the official website of the National Aeronautics and Space Administration (NASA) (https://ladsweb.modaps.eosdis.nasa.gov, accessed on 1 October 2022) [17]. Research [18] shows that internal ocean waves in the South China Sea occur frequently in summer, but less frequently in other seasons. Therefore, the MODIS data collection time used in this article is from 1 June to 31 August every year, and is conducted in the northern South China Sea. In order to reflect the data enhancement work, this article only collected a total of 1217 pieces of data from 2019 to 2022. Then, use ENVI Classic 5.3 professional softwareto read the HDF format file in Earth View 250M Reflective Solar Bands Scaled Integers format into BSQ format, then save it into IMG image format, and apply histogram equalization operation to improve the brightness of the image, where the maximum resolution. The rate is $5416 \times 8120$. The image is then split into smaller sub-images, each $64 \times 64$ pixels in size. Finally, experts who have studied remote sensing for many years divided these sub-images into different categories, including 700 images each of internal waves, clouds, oceans, and land. Figure 3 shows some classification results. The above process finally provides data support for this study, allowing us to study the existence and characteristics of internal waves in the ocean.

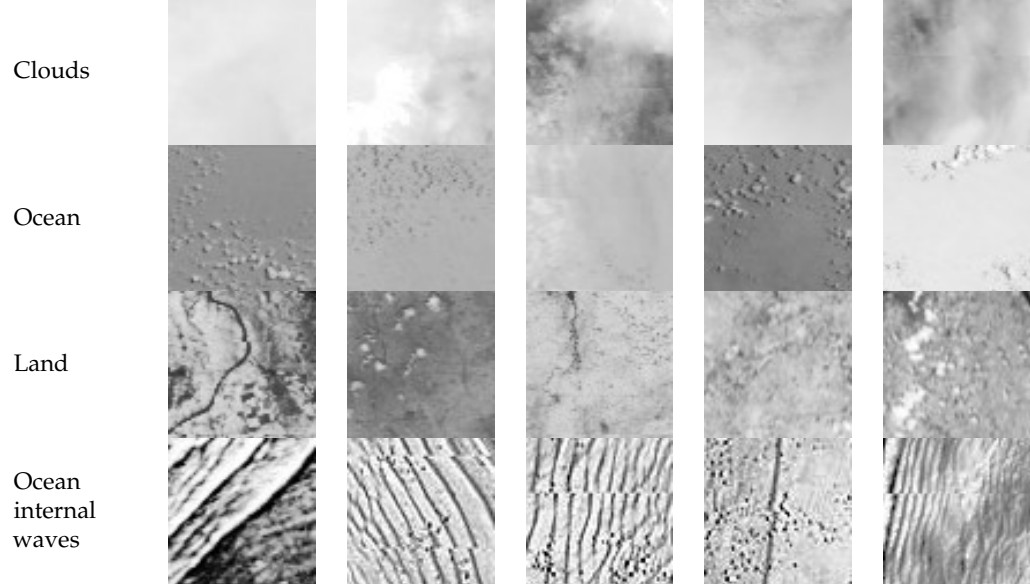

**Figure 3.** Partial data samples of clouds, ocean, land, and oceanic internal waves.

3.1.2. Data Augmentation

Currently, data augmentation techniques can be categorized into two main groups: traditional data augmentation methods and image generation algorithms based on generative adversarial networks (GANs), which are relatively new and capable of generating images with similar features to the original dataset but different from them. This approach significantly enhances dataset diversity, thereby improving the model's generalizability [19].

In order to compare the effects of these two data enhancement methods on training models, we applied these two methods to the original data and conducted experimental comparisons. First, we applied ten traditional data augmentation methods to each class of training samples. These methods include color truncation, min–max normalization, standard normalization, flipping, sharpening, Gaussian filtering, random erasing, random brightness transformation, random contrast transformation, and uniform noise. For each augmentation, we randomly selected 130 images, resulting in a total of 1300 augmented samples for each class of training data. When combined with the original dataset, this created a total of 2000 training samples for each class, constituting our Training Set 1.

Subsequently, we applied the GAN to data augmentation. Generative adversarial networks (GANs) are a type of deep learning model employed for generating synthetic data, including images [20]. This method leverages two neural networks: a generator and a discriminator. The generator's objective is to produce synthetic data that closely resembles real data by learning the distribution of real data [21]. Conversely, the discriminator is tasked with distinguishing between real and synthetic data. These two networks engage in a competitive process during training, leading to the continual improvement of the generator's ability to produce realistic synthetic data and the discriminator's ability to effectively differentiate between real and synthetic data.

The objective function $V(D; G)$ for the GAN is as follows:

$$min_G max_D V(D, G) = E_{X \sim P_{data(x)}}[log D(x)] E_{z \sim P_z(z)}[log(1 - D(G(z)))] \tag{1}$$

In the equations provided, where x represents a real sample, $D(x)$ signifies the probability assigned by the discriminator networks for classifying $x$ as a real sample. $G(z)$ corresponds to a sample generated from noise $z$ by the generator network $G$, and $D(G(z))$ indicates the probability assigned by the discriminator network $D$ for classifying $G(z)$ as a real sample.

Deep convolutional generative adversarial networks (DCGAN) [22] represent a variant of GANs designed to transform noise into images. They excel at generating images that fall within the same category as those present in the training set. DCGANs combine convolutional neural networks (CNNs) with unsupervised learning in the context of supervised learning, finding widespread applications in image generation.

In order to facilitate the generation of remote sensing images with dimensions of 64 × 64 pixels, we have made modifications to the DCGAN network architecture, as depicted in Figure 4. These modifications include the use of convolutional neural networks as both the generator and discriminator, the implementation of batch normalization for expedited training, the utilization of the LeakyReLU activation function to overcome the limitations of ReLU, the elimination of fully connected layers to prevent overfitting, the adoption of the Adam optimizer, and the capacity to generate high-quality images.

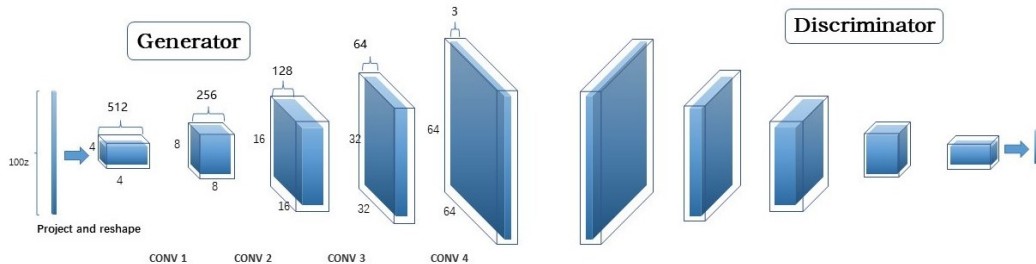

**Figure 4.** Modified DCGAN network structure.

We applied the modified DCGAN data augmentation to clouds, land, ocean, and internal waves, with parameters listed in Table 1.

**Table 1.** Experimental parameters for the modified DCGAN network.

| Original Data Volume | Batch Size | Learning Rate | Training Epochs | Optimizer | Exponential Decay Rate for the First Moment Is Estimated in the Optimizer | Exponential Decay Rate for the Second-Moment Estimates in the Optimizer |
|---|---|---|---|---|---|---|
| 700 | 16 | 0.0005 | 1000 | Adam | 0.5 | 0.999 |

The modified DCGAN network is trained using the original dataset, with the addition of a dropout layer at the end of each layer in the generator. This dropout layer serves to reduce the model's reliance on specific input features, enhance its generalization ability, and mitigate overfitting. The dropout parameter is set to 0.5, indicating that 50% of the neurons are randomly deactivated during each training iteration.

The training process for the modified DCGAN network has demonstrated success, as evidenced by the stabilization of loss functions for both the generator G and the discriminator D after 500 rounds. This suggests that the model has effectively learned the underlying data features and can generate images that closely resemble those in the original dataset.

Following this, the generator G is employed to generate 1300 images for each category, including ocean internal waves, clouds, land, and ocean. A selection of these generated images is presented in Figure 5. To further enhance the dataset, these generated images were combined with the original dataset, resulting in the creation of a new data augmentation training set referred to as Training Set 2.

The generated images, whether depicting clouds, oceans, land, or internal waves, convincingly simulate various real-world scenarios. These images effectively capture the texture of cloud layers, the topography of the land, the undulations of the ocean's surface, and the oscillations of internal waves. Consequently, our approach provides an effective means to expand existing remote sensing image datasets, with potential applications extending to various other applications within the realm of remote sensing. The dataset consisting of original data, traditional data augmentation, and the modified DCGAN-generated data are shown in Table 2.

In Part 4, we will compare the effects of two data enhancement methods in model training.

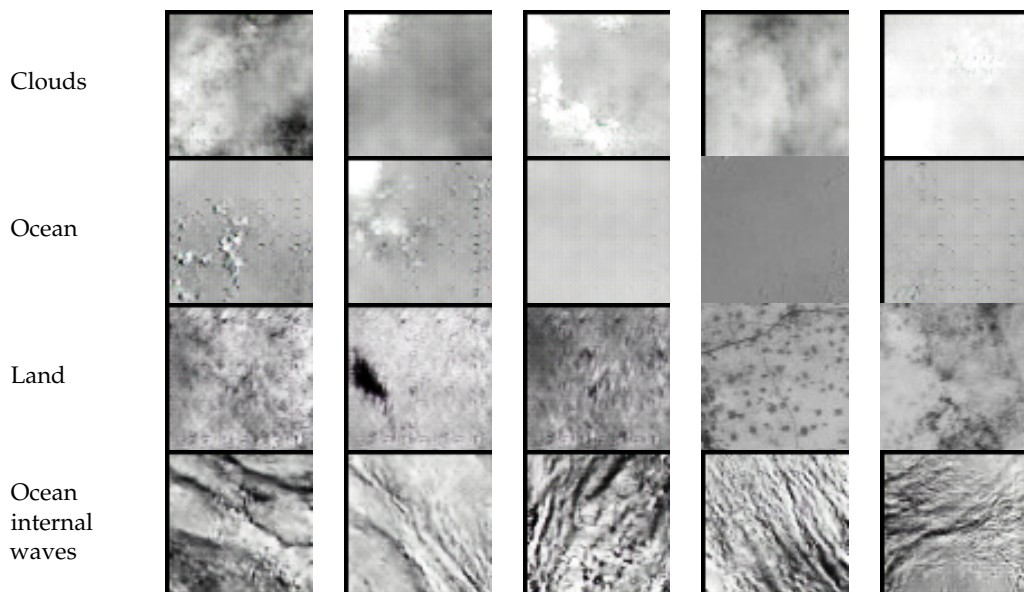

**Figure 5.** Some of the images generated by the modified DCGAN network.

**Table 2.** Number of training and testing sets.

| Unit: Images | Original Data Volume | Dataset Size for Traditional Data Augmentation | Dataset Size Generated by the Modified DCGAN | Training Set 1 | Training Set 2 | Test Set |
|---|---|---|---|---|---|---|
| Cloud | 700 | 1300 | 1300 | 2000 | 2000 | 400 |
| Land | 700 | 1300 | 1300 | 2000 | 2000 | 400 |
| Ocean | 700 | 1300 | 1300 | 2000 | 2000 | 400 |
| Oceanic internal waves | 700 | 1300 | 1300 | 2000 | 2000 | 400 |

### 3.2. Construction of the WaveNet Network Model

This article provides a detailed introduction to a residual convolutional neural network called "WaveNet," enhanced by a channel attention mechanism. WaveNet is designed to effectively process complex remote sensing images while achieving high classification accuracy and robustness. It achieves greater network depth by sequentially combining convolutional layers and pooling layers, enabling autonomous learning and the capture of essential features in remote sensing images. Simultaneously, it employs a channel attention mechanism to assign varying weights to channels within the feature map. These learned features are subsequently consolidated through a fully connected layer to produce the final classification result. The experimental findings presented in Section 4 unequivocally demonstrate the outstanding performance of WaveNet in diverse image classification tasks. Consequently, this method holds immense potential for widespread application in remote sensing image processing and is poised to contribute significantly to advancements in the field of remote sensing image classification.

### 3.2.1. Residual Block

The residual block structure in WaveNet is illustrated in Figure 6. The size of the input feature map is $C/2 \times H \times W$, where C represents the number of channels in the feature map, and H and W represent the height and width of the feature map, respectively. Between every two convolutional layers, there is a batch normalization layer and a rectified linear unit (ReLU) activation function [23]. The use of batch normalization ensures that the input distribution of each neuron remains consistent, which can accelerate the convergence speed of the network and avoid the issues of gradient vanishing and exploding, thereby improving the generalization performance of the model. ReLU is a commonly used activation function in deep learning. ReLU is defined as follows:

$$f(x) = max(0, x) \tag{2}$$

In short, for input $x$, if $x$ is greater than zero, then output $x$, otherwise output zero. The advantage of the ReLU activation function is its simplicity and non-linearity. Compared with traditional activation functions (such as sigmoid or tanh), ReLU is more stable for the back propagation of gradients and helps alleviate the vanishing gradient problem. In addition, ReLU introduces non-linearity, allowing the neural network to learn more complex functions.

In the architecture of the residual block within WaveNet, the initial convolutional layer plays a pivotal role. Its primary function is to double the number of channels in the feature map while halving its spatial size. This process is crucial for maintaining consistent feature map sizes during the initial addition operation. To achieve this, we employ a $1 \times 1$ convolutional layer that processes the input data, augmenting dimensionality along the channel axis and aligning it with the dimensions of other components within the residual block [24].

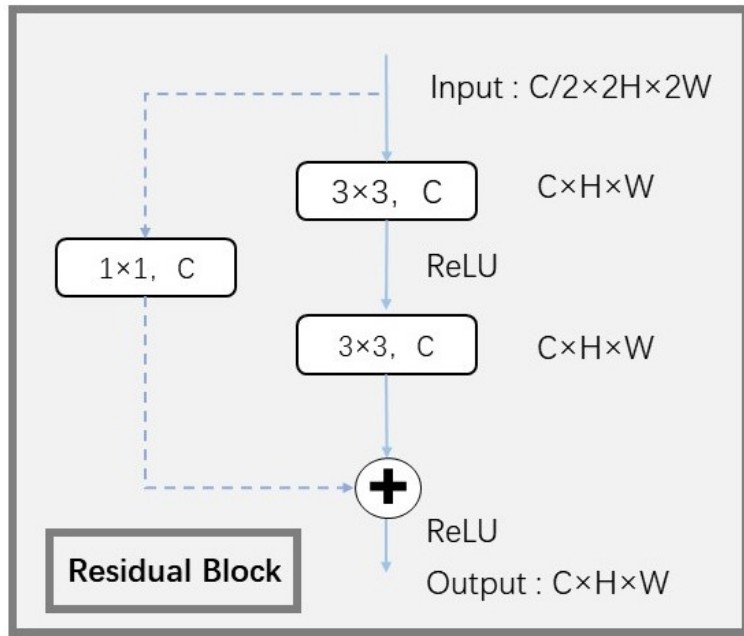

**Figure 6.** Residual block structure in WaveNet.

Following this dimension alignment, a $3 \times 3$ convolutional layer is utilized for feature extraction. Importantly, the resulting feature map maintains the same spatial dimensions as the input feature map. Subsequently, the output feature map from the initial convolutional layer undergoes further feature extraction via another $3 \times 3$ convolutional layer. An element-wise addition operation is then applied to the output feature map of the second convolutional layer, effectively creating a residual connection. This connection enables the network to learn the difference between the input and output, which enhances network optimization and training.

After the residual connection, we apply a ReLU activation function to the feature map, effectively setting all negative values to zero. This introduces non-linear features and amplifies the network's representational capacity. Through this sequence of operations, the residual block efficiently extracts and propagates essential feature information, thereby enhancing the network's overall performance and its ability to learn complex patterns within remote sensing images.

The mapping relationship in the residual block can be succinctly represented as follows:

$$y = F(x, W_i) + W_s x \tag{3}$$

In this context, we utilize the symbols x and y to represent the input and output feature maps of the residual block, respectively. The primary objective of the residual block is to acquire knowledge of the residual mapping function $F(x, W_i)$, with $W_i$ denoting the set of parameters involved in the learning process. To facilitate the integration of shortcut connections and ensure dimension alignment, a $1 \times 1$ convolutional layer is introduced, with parameters $W_s$ responsible for managing dimension adjustments. These pivotal steps within the residual blocks of WaveNet effectively facilitate the extraction of essential features and maintain stable gradient flow during training. Consequently, this simplifies the training of deep neural networks by addressing challenges related to vanishing and exploding gradients.

### 3.2.2. SE Residual Block

The squeeze and excitation (SE) residual block, illustrated in Figure 7, represents an essential architectural component within WaveNet. It combines the benefits of a traditional residual block with a channel-wise attention mechanism. This integration enhances the model's capability to capture and leverage crucial global information across feature chan-

nels, facilitating the recognition of important patterns and relationships within the data. The SE residual block plays a pivotal role in enhancing the overall performance of the WaveNet model.

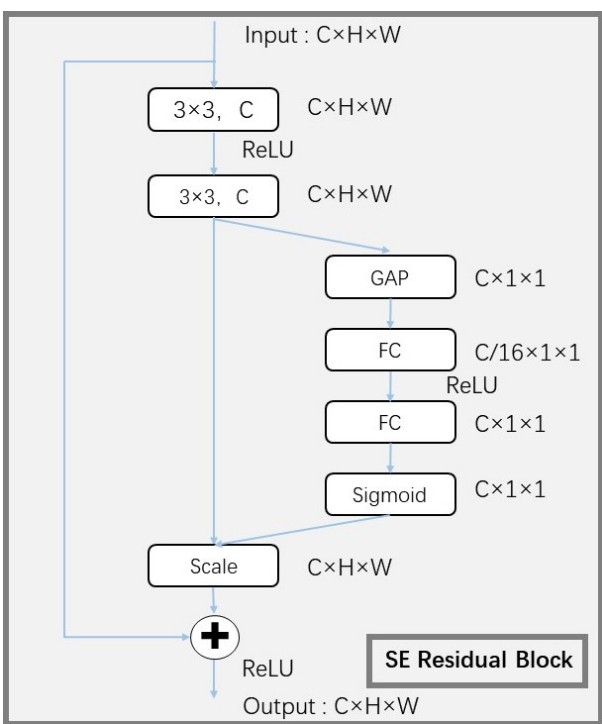

**Figure 7.** The squeeze and excitation attention mechanism in the residual block (SE residual block) structure.

The initial size of the input feature map $X_0$ of the SE residual block is C × H × W. and the output feature map X after two layers of convolution operations will be used as the input of global average pooling. We will perform feature compression on the feature map X along the spatial dimension by applying global average pooling. This operation transforms each two-dimensional feature channel into a single scalar value. Each scalar, in a sense, possesses a global receptive field and shares the same dimensionality as the number of input feature channels. These scalars represent the global response distribution across feature channels and enable layers closer to the input to access global information. Consequently, feature compression transforms the size of the feature map from C × H × W to C × 1 × 1. Following feature compression, the resulting compressed feature vector serves as the input to the channel-wise attention mechanism. This integrated mechanism effectively allows the model to capture and leverage global information across feature channels, thereby enhancing its capacity to recognize important patterns and relationships within the data.

The formula of global average pooling of the SE residual block can be expressed as follows:

$$F_c(X) = \frac{1}{H \times W} \sum_{i=1}^{H} \sum_{j=1}^{W} X(i,j) \tag{4}$$

In the equation, $F_c$ represents the compressed feature map, and $X$ represents the output feature map of the last convolutional layer.

The subsequent two fully connected layers are used to model the correlations between channels and output weights equal to the number of input feature channels. Firstly, in the first fully connected layer, we reduce the dimensionality of the channel features to 1/16 of the original size. Then, a ReLU activation function is applied for non-linear transformation, followed by another fully connected layer to increase the dimensionality back to the dimension of the original feature channels. This design, compared to using

only one fully connected layer, offers greater non-linear capability, enabling better fitting of complex correlations between channels, while also reducing the number of parameters and computational complexity. For the fully connected layer input FC, the output can be expressed as:

$$F_o = (ReLU(F_c \times w_1 + b_1)) \times w_2 + b_2 \tag{5}$$

In the equation, $F_o$ represents the final output of the fully connected layer, and $w_1$ and $b_1$ denote the weight and bias of the first fully connected layer, respectively. Similarly, $w_2$ and $b_2$ represent the weight and bias of the second fully connected layer.

The output from the final fully connected layer undergoes a non-linear transformation facilitated by a sigmoid activation function. The formula of the sigmoid function is as follows.

$$Sigmoid(x) = \frac{1}{1 + e^{-x}} \tag{6}$$

The sigmoid function generates a normalized weight for each channel within the range of 0 to 1. These normalized weights are then applied in an element-wise multiplication operation with the original channel features, thus finalizing the re-scaling of the original features along the channel dimension. This re-scaling process allows the network to assign weights to individual features based on their importance for the given classification task. Channels that are more relevant to the task receive higher weights, while the influence of less relevant channels is suppressed. By employing this approach, the network effectively harnesses the inter-channel correlations, elevates its feature representation capacity, and enhances overall classification performance. This mechanism ensures that the network pays greater attention to the most informative features, thereby improving its ability to recognize complex patterns and make accurate predictions. This process can be represented as follows:

$$\acute{X} = X \odot Sigmoid(F_o) \tag{7}$$

$\acute{X}$ represents the feature map after re-scaling. X denotes the output feature map of the last convolutional layer. $F_o$ represents the output of the last fully connected layer.

For the SE residual block, the size of the final output feature map Y is also C × H × W, which can be expressed by the following formula:

$$Y = ReLU(X_0 + X') \tag{8}$$

Among them, $X_0$ represents the input feature map of the SE residual block, $\acute{X}$ represents the feature map after re-scaling.

### 3.2.3. WaveNet

The WaveNet network architecture is composed of several key components, including convolutional layers with a 3 × 3 kernel size, max-pooling layers with a 2 × 2 size, three residual blocks, three channel-wise attention mechanism residual blocks, and a global average pooling layer. The final layer is a fully connected layer that utilizes softmax transformation to derive the probability distribution for each sample across different classes.

For input remote sensing images with dimensions of 3 × 64 × 64, each layer of the WaveNet network has specific input and output feature map sizes, as illustrated in Table 3.

To manage computational complexity effectively, WaveNet initially employs a sequence of 3 × 3 convolutional layers and 2 × 2 max-pooling layers to reduce the feature map size. However, in the first residual block, the feature map size remains unchanged to preserve crucial information, while the number of channels is doubled. In the subsequent three residual blocks, the channel count is doubled, but the height and width are halved by the first convolutional layer.

The final layer in the network is a fully connected layer that incorporates softmax transformation. The last layer in the network is a fully connected layer containing a softmax

transformation. This layer converts the final output of the network into a probability distribution between 0 and 1. The formula of the softmax function is as follows:

$$p_i = \frac{e^{z_i}}{\sum_{j=1}^{K} e^{z_j}} \qquad (9)$$

Among them, $p_i$ represents the probability of the $i$-th category, $z_i$ is the $i$-th element of the input vector $z$, and $K$ is the total number of categories, which is 4 in this article.

The softmax function has specific advantages in classification tasks because it can calculate the predicted probability for each class, which is very useful for the training and interpretation of neural networks. The softmax function will emphasize the element with the largest value in the input vector so that its corresponding category probability is the highest, thereby classifying, and ultimately determines the final classification result.

**Table 3.** Size changes of feature map.

| Network Layer | Input Feature Map Size | Output Feature Map Size |
|---|---|---|
| 3 × 3 Conv | 3 × 64 × 64 | 64 × 64 × 64 |
| 2 × 2 Max Pooling | 64 × 64 × 64 | 64 × 32 × 32 |
| Residual Block | 64 × 32 × 32 | 128 × 32 × 32 |
| SE Residual Block | 128 × 32 × 32 | 128 × 32 × 32 |
| Residual Block | 128 × 32 × 32 | 256 × 16 × 16 |
| SE Residual Block | 256 × 16 × 16 | 256 × 16 × 16 |
| Residual Block | 256 × 16 × 16 | 512 × 8 × 8 |
| SE Residual Block | 512 × 8 × 8 | 512 × 8 × 8 |
| Global Average Pooling | 512 × 8 × 8 | 512 × 1 × 1 |
| Fully Connected | 512 | 4 |

### 3.3. Transfer Learning

In the domain of deep learning models, training models with numerous hidden layers and parameters often demand a substantial amount of high-quality labeled data, which can result in significant time and computational costs. However, when dealing with satellite imagery of oceanic internal waves, data availability is frequently limited, presenting a challenge in training an effective deep learning model with a small dataset. To address this limitation, transfer learning has emerged as a valuable approach in deep learning model training, alleviating the data requirement.

Transfer learning involves leveraging a pre-trained model, which is then fine-tuned for a new task. The fundamental concept of transfer learning is to initially train the network model parameters using large-scale datasets and subsequently fine-tune them for the specific image recognition task at hand. This approach greatly assists the classifier in performing image recognition tasks, even when confronted with limited data resources.

The EuroSAT remote sensing dataset, as described in reference [25], comprises 10 distinct scene categories, including agricultural land, forest, herbaceous vegetation, highways, industrial areas, pastures, permanent crops, residential areas, rivers, and lakes, with a total of 3000 samples for each category. The primary objective of this study is to evaluate the performance of employing the pre-trained WaveNet network for classification tasks using the EuroSAT remote sensing dataset. Detailed parameter settings for this study are provided in Table 4. These settings cover various aspects of the model and training process, offering a comprehensive overview of the experimental configuration and methodology employed in the classification task using the EuroSAT dataset.

**Table 4.** Pre-trained network parameters.

| Batch Size | Learning Rate | Training Epochs | Optimizer |
|---|---|---|---|
| 32 | 0.001 | 500 | Adam |

During the fine-tuning stage, the parameters of the WaveNet network model are initialized with pre-trained parameters, and the output of the last fully connected layer is adjusted to 4 in order to perform classification on the MODIS image dataset. Utilizing a pre-training strategy instead of initializing the network parameters with random weights allows the WaveNet model to initially learn rich texture features from the EuroSAT dataset. This approach not only eliminates the need to train the model from scratch but also helps overcome potential overfitting issues, thereby enhancing the model's performance in situations with limited data. By leveraging pre-training, the model can benefit from the learned representations, enabling it to generalize more effectively and achieve higher performance even when the available data are limited.

## 4. Experiments and Results

In this paper, we will use PyTorch as a deep learning framework to build network models. We also utilize graphics processing units (GPUs) to improve computing speed and training efficiency during network training.

In the experimental phase, we will first analyze the quality of data generated based on the modified DCGAN through the t-SNE dimensionality reduction method. Then, compare the training effects of Datasets 1 and 2 obtained by using two different data enhancement methods on the WaveNet network. Subsequently, we will compare the training effects of the WaveNet network using the transfer learning strategy and not using the transfer learning strategy. Finally, we compare the results of WaveNet with previous related work.

Table 5 outlines the hardware and software environment during the experiment.

**Table 5.** Parameters of experimental conditions.

| Hardware Equipment | Software Environment |
|---|---|
| CPU: Intel(R) Xeon(R) Gold 5218R 2.10 GHz | Rocky Linux 8 |
| RAM:32GB | CUDA 11.4 |
| GPU: NVIDIA RTX 3090 | Pytorch 1.12.1 |

### 4.1. Analysis of Generated Images Using the Modified DCGAN

In order to analyze the data distribution generated by the modified DCGAN more intuitively, the t-distributed stochastic neighbor embedding (t-SNE) [26] dimensionality reduction method is used in this paper. t-SNE is an algorithm for visualizing high-dimensional datasets by measuring the distance between each data point and other data points to calculate their correlation [27]. We randomly select 200 images from each category of the original data and the modified DCGAN-generated data for testing (as shown in Figure 8), where different colors represent different labels.

The results of t-SNE dimensionality reduction highlight the effectiveness of the modified DCGAN in generating synthetic remote sensing images that closely resemble real-world images while introducing valuable diversity. The key takeaways from this research include:

1. Realism and Diversity: The modified DCGAN has demonstrated its capability to produce synthetic remote sensing images that exhibit realism, making them highly similar to authentic remote sensing data. Additionally, the diversity observed within the generated images is a significant advantage. The ability to generate diverse samples within each class contributes to a more comprehensive and representative dataset.

2. Matching Feature Distribution: The use of t-SNE for dimensionality reduction and visualization has substantiated that the feature distribution within the generated images closely aligns with that of real images. This alignment suggests that the modified DCGAN has successfully captured and retained essential features present in real remote sensing data, enhancing the data's quality.

3. Enhanced Data for Training: By effectively generating realistic and diverse data samples, the modified DCGAN provides a valuable resource for training neural network

models. This expanded dataset can be instrumental in improving the generalization capabilities of recognition models, as it exposes them to a wider range of scenarios and variations present in the real world.

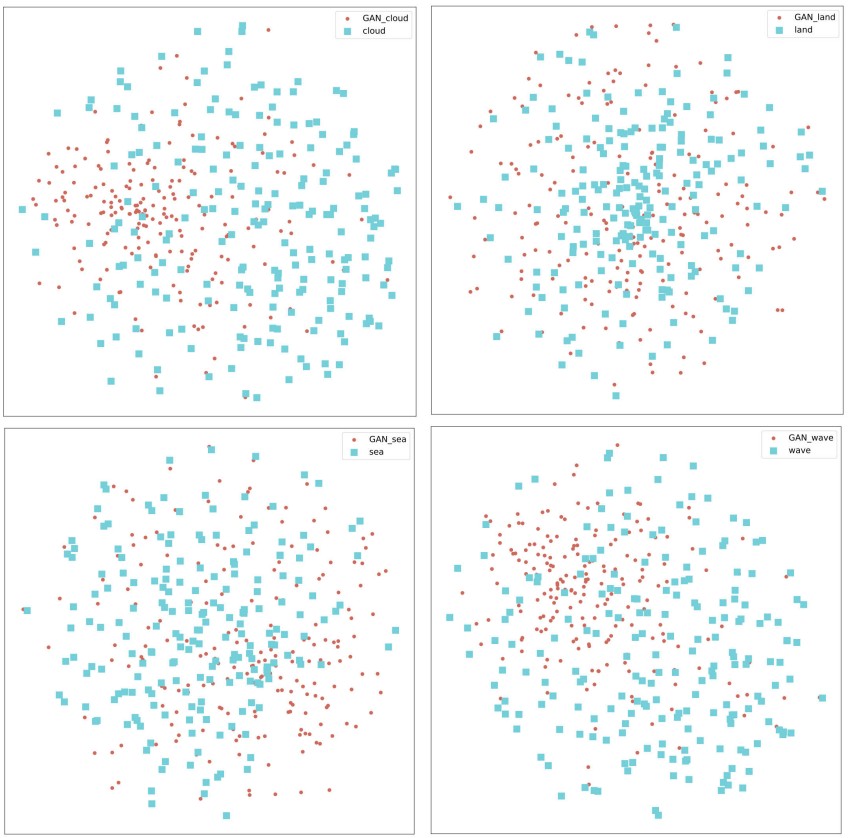

**Figure 8.** T-SNE dimensionality reduction is applied to different data augmentation datasets, where red represents the data generated by the modified DCGAN and blue represents the original data.

*4.2. Comparison of Classification Results of the WaveNet Network Using Different Training Sets*

This article evaluates the WaveNet model using the overall accuracy (OA) and average accuracy (AA) metrics for classification. The confusion matrix $S$ is a $L \times L$ matrix, $L$ that represents the number of classes. $S_{ij}$ represents the number of test samples that belong to the class i and were classified as a class j. The total number of test samples is $M = \sum_i^L \sum_j^L S_{i,j}$.

The overall accuracy (OA) metric provides a good description of the overall classification accuracy, where OA is calculated by dividing the number of correctly classified samples by the total number of test samples, which can be represented as follows:

$$\text{OA} = \frac{\sum_i^L S_{i,j}}{M} \times 100\% \tag{10}$$

The average accuracy (AA) metric provides a good description of the classification performance differences among each class, representing the average classification accuracy for each class, which can be represented as follows:

$$\text{AA} = \frac{\sum_i^L \left( \frac{S_{i,j}}{\sum_j^L S_{i,j}} \right)}{L} \times 100\% \tag{11}$$

Precision is a metric widely used to evaluate the performance of classification models. It measures the accuracy of the model in predicting positive examples. The calculation formula of accuracy is as follows:

$$Precision_i = \frac{S_{i,i}}{\sum_j^L S_{j,i}} \times 100\% \qquad (12)$$

In this formula, $Precision_i$ represents the precision for class $i$. $S_{i,i}$ denotes the number of samples correctly classified as class i, and $\sum_j^L S_{j,i}$ represents the total number of samples predicted as class $i$.

Recall is also one of the indicators widely used to evaluate the performance of classification models. It measures the proportion of actual positive examples identified by the model, that is, how many of all actual positive examples were correctly predicted as positive by the model. The calculation formula of recall rate is as follows:

$$Recall_i = \frac{S_{i,i}}{\sum_j^L S_{i,j}} \times 100\% \qquad (13)$$

In this formula, $Recall_i$ represents the recall for class $i$. $S_{i,i}$ denotes the number of samples correctly classified as class $i$, and $\sum_j^L S_{i,j}$ represents the total number of samples actually belonging to class $i$.

To investigate whether the images generated by the modified DCGAN network possess features similar to those of real images and whether they are more effective as a data augmentation technique than traditional methods, enhancing the network's generalization ability, we conducted a series of comparative experiments. The overall accuracy (OA) and average accuracy (AA) experimental results are shown in Table 6. The precision of each category is shown in Table 7. The recall rates of each category are shown in Table 8.

**Table 6.** Accuracy results of WaveNet under the same experimental parameters using training sets based on traditional data augmentation and DCGAN-generated data augmentation.

|  | **Traditional Data Augmentation/%** | **Data Augmentation Based on the Modified DCGAN/%** |
|---|---|---|
| Overall accuracy | 93.188 | 98.625 |
| Accuracy of cloud recognition | 97.000 | 99.750 |
| Accuracy of land recognition | 90.750 | 97.250 |
| Accuracy of ocean recognition | 90.250 | 98.500 |
| Accuracy of oceanic internal waves recognition | 94.750 | 99.000 |

**Table 7.** Results of precision of WaveNet under the same experimental parameters using a training set based on traditional data augmentation and DCGAN-generated data augmentation.

|  | **Traditional Data Augmentation/%** | **Data Augmentation Based on the Modified DCGAN/%** |
|---|---|---|
| Precision of cloud recognition | 92.38 | 98.28 |
| Precision of land recognition | 91.44 | 98.73 |
| Precision of ocean recognition | 91.62 | 98.25 |
| Precision of oceanic internal waves recognition | 97.43 | 99.25 |

By using the modified DCGAN data augmentation method proposed in this paper, we observed an overall improvement of 5.437% in the classification accuracy of the test set, with all four types of remote sensing images showing increased recognition rates. Specifically, the recognition accuracy of clouds, land, oceans and internal waves improved by 2.75%, 6.5%, 8.25%, and 4.25%, respectively. Thanks to the modified DCGAN data

enhancement method proposed in this article, the WaveNet model has also been greatly improved on the test set in terms of precision and recall indicators of each category. For example, the recognition accuracy of land increased by 7.29%, while the internal accuracy of identifying waves increased by 1.82%; the recall rate of identifying oceans increased by 8.25%; and the recall rate of identifying internal waves also increased by 4.25%.

**Table 8.** Results of recall of WaveNet under the same experimental parameters using a training set based on traditional data augmentation and DCGAN-generated data augmentation.

|  | Traditional Data Augmentation/% | Data Augmentation Based on the Modified DCGAN/% |
| --- | --- | --- |
| Recall of cloud recognition | 97.00 | 99.75 |
| Recall of land recognition | 90.75 | 97.25 |
| Recall of ocean recognition | 90.25 | 98.50 |
| Recall of oceanic internal waves recognition | 94.75 | 99.00 |

### *4.3. Discussion*

For fair comparison, all works are pre-trained on the EuroSAT dataset. And the hyperparameter settings are as shown in Table 9.This article shows the performance comparison results of each model on the test set in Table 10. Our proposed WaveNet achieves the highest accuracy, which further illustrates the effectiveness of the WaveNet model. Other models have a high number of layers and a large number of parameters, but their performance is poor. This may be because the larger the model, the larger the dataset required for learning to ensure that the model is effective.

Figure 9 shows the t-SNE 2D visualization of semantic features extracted by WaveNet on datasets enhanced by traditional data augmentation and data augmentation based on the modified DCGAN, respectively. The feature distribution of the same type of data has a large overlap, while the data of different types are far apart. The feature distribution of each type of image shows an obvious balloon-like distribution, which shows that for WaveNet, these subtle features are distinguishable.

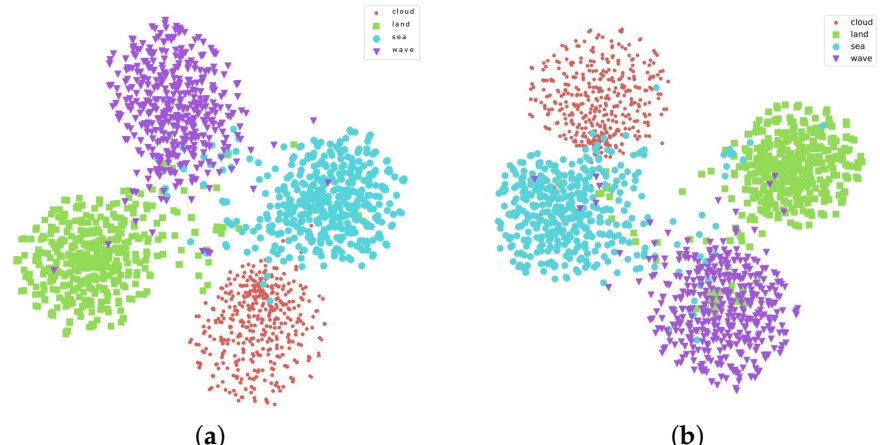

(**a**)          (**b**)

**Figure 9.** Classification results of WaveNet using different data augmentation methods. (**a**) Classification results of WaveNet using the DCGAN-augmented dataset. (**b**) Classification results of WaveNet using the traditionally augmented dataset.

**Table 9.** Hyperparameters for training.

| Batch Size | Learning Rate | Training Epochs | Optimizer |
| --- | --- | --- | --- |
| 128 | 0.001 | 500 | Adam |

**Table 10.** Performance comparison on the dataset.

| Methods | Overall Accuracy/% | Accuracy of Oceanic Internal Waves Recognition/% |
| --- | --- | --- |
| AlexNet [28] | 95.00 | 95.25 |
| VGG11 [29] | 97.875 | 96.75 |
| VGG16 [29] | 97.50 | 96.25 |
| GoogLeNet [30] | 98.50 | 98.75 |
| Resnet18 [24] | 98.125 | 98.75 |
| WaveNet (ours) | 98.625 | 99.00 |

*4.4. Display of Test Results*

The results shown in Figure 10 demonstrate that by cropping the original remote sensing images and inputting them into the WaveNet network for classification, the ocean internal waves in the MODIS images were successfully identified with high accuracy. This is a notable achievement as accurately identifying oceanic internal waves is a challenging task in remote sensing image analysis.

By outlining the areas classified as "internal waves" in the original image, the figure provides a visual representation of the algorithm's effectiveness in detecting these features. This successful identification of oceanic internal waves has practical implications for various applications, including oceanography and environmental monitoring, where the detection and tracking of such phenomena are crucial.

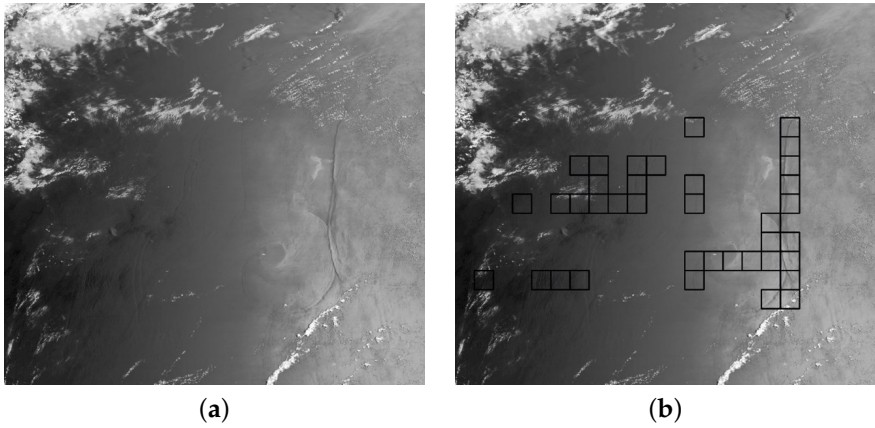

(**a**)　　　　　　　　　　　　　　(**b**)

**Figure 10.** Display of test results: (**a**) a whole remote sensing image; (**b**) remote sensing images after being detected.

**5. Conclusions**

This paper uses MODIS remote sensing data to produce different types of remote sensing image samples, including internal waves, processes these images, and generates a database including internal waves. In addition, this paper proposes an end-to-end method that uses deep learning technology to improve the recognition performance of MODIS remote sensing images. One of the key contributions is the use of a modified DCGAN network for data augmentation, which significantly improves the diversity of the dataset and enhances the generalization ability of the recognition model. This method has the following advantages: (1) Reduce data collection costs: collecting and labeling large-scale remote sensing datasets is expensive and time-consuming. By using the modified DCGAN to generate data, the time and resources for collecting real data can be effectively reduced; (2) Increase data diversity: The data generated by the modified DCGAN are very similar to real remote sensing images. The data it generates contain a wider range of variations and scenarios, helping network models learn and process more complex real-world data more effectively; (3) Improve generalization ability: The combination of real data and synthetic data improves the generalization ability of the model. This means that the trained model is more able to accurately classify and identify new, unknown data.

Another contribution lies in the design of the WaveNet network with excellent recognition performance. By adding a channel attention mechanism to the deep convolutional layer, WaveNet can pay different attention to each channel of the feature map, thereby more effectively learning features related to remote sensing classification. Moreover, the convolutional layers of WaveNet have a residual structure, which allows WaveNet to avoid overfitting problems caused by too deep layers in actual training.

By combining the above methods, we have improved the recognition accuracy of remote sensing image classification tasks. This has practical applications in many fields, such as environmental monitoring, early warning of marine disasters, detection of internal ocean waves, etc. However, due to the fact that the data come from optical remote sensing satellites, it may be difficult to obtain sea surface data when the weather is bad, resulting in certain limitations in actual citation.

In future research, we plan to further explore how to integrate geographical location information into remote sensing images to further improve the practicality of the research. Specifically, we plan to collect data on the geographical coordinates of where remote sensing images were taken and associate these data with the images. Through this association we can achieve the following goals.

Geographical information feature extraction: We plan to use geolocation information to extract information related to geographical features in images. For example, we can determine the distance of waves within the ocean from the shoreline in an image, as well as the shape and wavelength of the waves. This information is of great significance to fields such as ocean research.

Environmental monitoring and management: Geolocation information can also be used for environmental monitoring and management. We plan to use this information to track changes in specific areas, such as changes in land use or changes in ocean water quality. This will lead to a better understanding and management of natural resources.

Geographic Information System (GIS): We also plan to use remote sensing images in conjunction with GIS technology to create a geographic information system. This will enable users to better visualize and analyze geographic data and support a variety of applications, from urban planning to natural disaster management.

These works will help apply remote sensing images to a wider range of fields and improve the practicality and adaptability of models.

**Author Contributions:** Methodology, Z.J. and X.G.; Project administration, Z.J.; Resources, Z.J., L.S., and N.L.; Writing—original draft preparation, X.G.; Writing—review and editing, L.S. and L.Z. All authors have read and agreed to the published version of the manuscript.

**Funding:** The authors gratefully acknowledge the co-funding and support of the work by the Jiangsu Key Research and Development Plan (BE2021012-2 and BE2021012-5), the Changzhou Science and Technology Support Program (CE20225034), and the Digital Twin Technology Engineering Research Center for petrochemical process support program (DTEC202002 ).

**Institutional Review Board Statement:** Not applicable.

**Informed Consent Statement:** Not applicable.

**Data Availability Statement:** The data that support the findings of this study are available from the corresponding author on reasonable request.

**Conflicts of Interest:** The authors declare no conflicts of interest.

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
