# Peer review of "Detection of Ocean Internal Waves Based on Modified Deep Convolutional Generative Adversarial Network and WaveNet in Moderate Resolution Imaging Spectroradiometer Images"

_applsci, doi:10.3390/app132011235_

Round 1

Reviewer 1 Report

Dear Author/s,
I have provided my opinions on the article separately for each section. 

Abstract

It might be beneficial to briefly mention the significance of the study in the context of global climate change to provide readers with a sense of the broader implications of the research.

The mention of t-SNE technology is good, but it might be beneficial to briefly explain its role or significance in the methodology to make the abstract more self-contained.

The abstract might benefit from a sentence that briefly discusses the limitations of the study, which would give readers a more rounded view of the research.

1. Introduction

1.1Internal Waves

This section provides a detailed background on internal waves and their significance, which sets the stage well for the rest of the paper. The transition between the discussion of MODIS and the characteristics of internal waves in satellite imagery seems a bit abrupt. It might be beneficial to add a sentence or two that bridges these topics more smoothly. The section might benefit from a brief mention of the specific challenges or gaps in existing research that this study aims to address, to provide a clearer rationale for the research.

2. Related Work

It might be beneficial to include a few more recent studies (2022-2023) to give readers a sense of the most current developments in the field.

The section could benefit from a brief discussion of the limitations or gaps in existing research, which would help to set the stage for the introduction of the new methodology in the next section.It might be beneficial to briefly mention how this study builds upon or differs from the studies mentioned in this section to provide a clearer sense of the study's novelty and significance.

3. Methods and Material

3.1 Methodology

It might be beneficial to elaborate a bit more on how the WaveNet model works, especially focusing on the channel-wise attention mechanism and how it contributes to improving classification accuracy.

Mentioning the specific modifications made to the traditional DCGAN and how it contributes to the study would add depth to this section.

Including a brief explanation or reference for the transfer learning method would be beneficial to give the reader a more rounded understanding of the methodology.

3.2 Data Augmentation

3.2.1 MODIS Data

In this section, including statistical details or characteristics of the data (like the number of images used, resolution, etc.) might provide a more comprehensive view. Mentioning why the specific time frame (June 1 to August 31 from 2019 to 2022) was chosen would give more insight into the data collection process.

3.2.2 Data Augmentation

In this section, it would be beneficial to explain how the generated images from DCGAN were validated to ensure they are suitable for training the model.

3.3 Construction of the WaveNet Network Model

In this section, including a brief explanation for terms like "ReLU activation function" and "batch normalization" would make the section more accessible to readers unfamiliar with these terms. Including a brief explanation of how the channel-wise attention mechanism improves the model's performance would be beneficial.

3.3.1 Residual Block

Including a brief explanation of how the residual block structure contributes to the overall performance of the WaveNet model would provide more depth.

3.3.2 SE Residual Block

Including a brief explanation of how the SE Residual Block improves the model's performance would be beneficial. Including a brief explanation or reference for terms like "global average pooling" and "sigmoid activation function" would make the section more accessible to readers unfamiliar with these terms.

3.3.3 WaveNet

Mentioning the specific advantages of using softmax transformation in the final layer would be beneficial.

3.4 Transfer Learning

3.4.1 Pre-training of the Network Model

Mentioning the specific advantages of using the EuroSAT dataset for pre-training would be beneficial.

4. Experiments and Results:

While you have described the experimental details in section 4.1, it might be beneficial to expand this section to include more information about the methodology. This could involve detailing the dataset used, the preprocessing steps, and a more in-depth explanation of the traditional and modified DCGAN data augmentation techniques.

Discussion Section (Missing): After section 4.4, it would be beneficial to include a discussion section where you interpret the results, compare them with previous studies, and discuss the implications of your findings.

5. Conclusions

In this section, it could benefit from a more detailed discussion of the implications of the findings, limitations of the study, and potential future directions in a broader context.

The future work section is well outlined, but it might be beneficial to expand on how incorporating geographic location information could potentially enhance the model's performance and the specific methodologies planned to achieve this.

The authors should proofread the document for any grammatical or typographical errors before submission.

Reviewer 2 Report

This paper examined a GAN approach for data augmentation to improve the WaveNet classification model, which is an interesting effort but requires modifications regarding data processing and model tests.

L155. MOD02QKM is the Level 1B calibrated radiances. How did you classify the radiance images into clouds, ocean, land, and internal waves? You just mentioned, “These images were categorized.” Please describe how you decided on the categories for these images. A specific explanation is needed.

L203. How did you decide that the hyperparameters were optimal? Please describe the optimization process in detail.

L220. Parts of the generated images in Figure 5 did not seem appropriate. For example, the third and fifth images for Ocean do not look like the other ocean images. We cannot make sure that the fifth image for Land belongs to Land or another class. How can you ensure the generated images are appropriate for each class?

L459. You should present a contingency table for the WaveNet test. Also, precision and recall should be presented, in addition to accuracy.

Round 2

Reviewer 1 Report

The authors have tried to make corrections by taking into consideration most of the criticisms. I believe that the article will have a widespread impact and contribute to science.

Author Response

Dear Reviewer,

Thank you very much for your review of our research article and for your valuable feedback. We greatly appreciate your suggestions and are pleased to hear that you believe we have adequately addressed most of the criticisms.

We have worked diligently to ensure the quality and scientific significance of the article, and we are encouraged by your confidence that it will make a substantial contribution to science. Your support motivates us to delve deeper into our research and to ensure its impact is as widespread as possible.

We are truly grateful for your professional insights and endorsement, which are crucial to our work. We have taken your suggestions on board and continue to improve the article to meet high standards. If you have any further comments or recommendations, we would be more than willing to consider them and make necessary revisions.

Once again, thank you for your time and patience, and we look forward to sharing the progress of this research with you. If you have any further questions or require additional information, please do not hesitate to let us know.

Warm regards,

Mr Gao